# Nerve injury induces pain hypersensitivity and anxiety-related behaviours and is associated with amygdala activation in male mice

Siyuan Tong ◍, Yuerong Chen◍, Zonglin Wu◍, Linbao Wang◍, Yuxin Wei, Xirui Wang, Yuanyuan Wu◍*

Key Laboratory of Acupuncture and Neurology of Zhejiang Province, Department of Neurobiology and Acupuncture Research, The Third Clinical Medical College, Zhejiang Chinese Medical University, Hangzhou, China

◍ These authors contributed equally to this work.
* runnaway@126.com

## Abstract

### Background

Neuropathic pain is a common health problem, often accompanied by anxiety. The amygdala plays a crucial role in emotional processing and the basolateral amygdala (BLA) and the central amygdala (CeA) are primary components of the amygdala. The c-Fos is considered indicative of neuronal activation. We studied whether BLA and CeA are activated and their internal activation when chronic neuralgia occurs.

### Methods

The spared nerve injury (SNI) model was employed for this investigation. Mechanical paw withdrawal thresholds (PWTs) were utilized for assessing pain hypersensitivity, followed by observation of anxiety-like behaviors using the elevated plus maze test (EPMT) and the open field test (OFT). The c-Fos in BLA and CeA were measured by immunofluorescence staining.

### Results

We found that SNI mice exhibit pain hypersensitivity and anxiety-like behaviors. The expression of c-Fos in the BLA and CeA was upregulated in SNI mice. Besides, there were spatial differences in c-Fos expression between BLA and CeA.

### Conclusion

The BLA and CeA showed activation in chronic pain and associated anxiety, and there were spatial differences in this activation. Targeting the amygdala may hold promise for treating chronic nerve pain with anxiety-like behaviors.

**Data availability statement:** All relevant data are within the manuscript and its Supporting information files.

**Funding:** The National Natural Science Foundation of China (NO. 8207151734), the Natural Science Foundation of Zhejiang Province (NO. LY23H270009).

**Competing interests:** The authors have declared that no competing interests exist.

## Introduction

Neuropathic pain, stemming from injury or disease affecting the somatosensory system, represents a particularly challenging form of chronic pain [1]. The management of neuropathic pain continues to present difficulties due to the limited efficacy of traditional pharmacological treatments, numerous side effects, and high rates of recurrence [2]. Along with the physical suffering, individuals experiencing neuropathic pain often experience accompanying negative emotional states such as depression and anxiety [3]. These emotional responses not only intensify the patient's perception of pain but also significantly impact their quality of life, social functioning, and mental health [4]. Anxiety and depression may cause patients to focus more on their pain, creating a vicious cycle of pain-emotion-pain, further exacerbating the pain experience [5–8].

The amygdala, a pivotal structure nestled deep within the temporal lobe of the brain, proximal to the hippocampus, constitutes a fundamental component of the limbic system [9–11]. Renowned as the "emotional brain", it wields immense influence over emotional regulation, learning and memory, processing of fear and anxiety, as well as social behaviors [12–15]. In addition to play a crucial role in emotional processing, the amygdala is often activated during the experience of pain [16]. Pain is not merely a sensory experience, but also accompanied by intense emotional components. The amygdala integrates pain signals from various parts of the body along with emotional information, forming a comprehensive response to pain [17]. Therefore, the amygdala's intricate interplay with pain and anxiety underscores its importance in modulating emotional responses and maintaining emotional homeostasis.

The basolateral amygdala (BLA) and the central amygdala (CeA) are primary components of the amygdala, with BLA consisting of approximately 80% glutamatergic excitatory neurons and about 20% GABAergic (γ-aminobutyric acid) inhibitory neurons [18], while CeA consists mainly of around 95% GABAergic neurons [19]. After neuropathic pain is induced, the CeA and the BLA undergo neuroplastic changes and exhibit enhanced stimulus-induced activity associated with tactile dyskinesia [20–22]. Studies have shown that when confronted with danger, the BLA activates specific neural pathways, prompting individuals to initiate active avoidance behaviors rather than passive immobility [23]. There are complex interactions between CeA and BLA, which jointly regulate individual defense behavior. When the BLA senses a threat and activates, it may promote harm avoidance by activating specific neurons in the CeA.

Studies have demonstrated that amygdala is implicated in anxiety-like behavior induced by spared nerve injury (SNI) [24]. The amygdala, as a structurally complex and functionally diverse brain region, exhibits distinct anatomical structures and functional subdivisions across different dimensions. Through in-depth exploration, its fine architecture can be analyzed from multiple perspectives, which delineate the amygdala's internal organization. These multifaceted investigations further provide critical insights into its involvement in higher cognitive functions, particularly emotional processing, memory consolidation, and social behavior. However, the

differential involvement of BLA and CeA along their rostrocaudal axis in SNI-induced neuropathic pain remain to be fully elucidated.

In this study, our objective is to investigate how rostro-caudal involvement of BLA and CeA when anxiety-like behaviors occur. The SNI model was employed for this investigation. Mechanical paw withdrawal thresholds (PWTs) were utilized for assessing pain hypersensitivity, followed by observation of anxiety-like behaviors using the elevated plus maze test (EPMT) and the open field test (OFT). The expression of c-Fos is considered indicative of neuronal activation and can be used to assess changes in excitability within brain nuclei such as the amygdala [25]. We analyzed the density of c-Fos by immunofluorescence staining at varying distances from the bregma.

## Materials and methods

### Experimental animals

For this study, male C57BL/6 mice (aged 6–8 weeks) were utilized. All the mice in the experiment were from the Experimental Animal Center of Zhejiang Chinese Medical University, having received official approval from the Association for Assessment and Accreditation of Laboratory Animal Care. After random allocation, the mice were randomly divided into 2 groups. Four mice were housed per cage, with corn cobs provided for bedding. The mice were housed in an environment equipped with a ventilation and air filtration system, as well as a temperature control system maintaining a temperature range of 23–25°C. Additionally, there was a standard 12-hour light/dark cycle, as well as an ample supply of food and water.

### Ethics approval

The experimental procedures adhered to the ethical requirements for experimentation set forth by the Experimental Animal Management and Ethics Committee of Zhejiang Chinese Medical University (approval number: IACUC-20240226–05).

### Animal model of neuropathic pain

Mice were anesthetized by inhalation of isoflurane (1%-3%). The middle left hind limbs of mice of the SNI group were shaved and disinfected. Then, a one-centimeter incision was made along the thigh to bluntly separate the underlying muscles and expose the sciatic nerve and its branches. Ligated the sural and common peroneal branches with 6–0 sutures and transect them. Simultaneously, the tibial nerve was kept unwounded and avoided being touched. Finally, the incisions were closed and disinfected layer by layer. For the sham group, mice underwent the same procedure but without nerve ligation or transection.

### Von Frey filament test

For this study, Von Frey filaments were used to measure the mechanical PWTs on the left hind paw of each mouse. At the same time, to facilitate the experiment, the mice were individually placed in a plexiglass chamber, which was on the wire-mesh platform. The mice were given 1 hour to acclimate to the chamber. After that, filament probes were inserted onto their left hind paws, with gradually increasing pressure. A positive mechanical withdrawal response was defined as the mice suddenly retracting their paws, licking their claws, or flinching. With intervals of more than 1 minute between each measurement, the PWT was determined as the pressure required to elicit 3 positive responses out of 5 stimuli. We measured the PWTs on different days, including the baseline measurement, day 7, and day 14.

### Elevated plus maze (EPM) test

The study used EPM test to assess anxiety-like behavior of the mice. The EPM apparatus, which consists of two opposite open arms, two opposite closed arms, and a central region, was 50 cm high from the ground. At the beginning of

the test, each mouse was placed in the central area facing the open arm and was allowed to explore the apparatus for 5 minutes. The camera analysis device, positioned directly above the maze, recorded the behavior of the mice, including the time spent in the open arms, the total number of open and closed arm entries (representing exploratory activity), and the frequency of head dipping in the open arms. All these data were used to calculate an anxiety index. Before placing each mouse in the maze, 70% alcohol was sprayed to eliminate any odor left by the previous mouse, thereby ensuring the reliability of the test.

### Open field (OF) test

The open field test was utilized to analyze the behaviors of the mice. We individually placed mice in the center of the field, which was constructed of acrylic panels and comprised individual activity chambers(50 × 50 × 25 cm). A camera was positioned directly above the field and used in conjunction with Smart3.0 software for recording and analysis. The analyzer automatically tracked the behavior trajectory of the mice, including the total distance traveled, the number of center crossings, and the distance from the center. Each mouse was observed for 5 minutes. To eliminate any impact of residual odor from the previous mouse on the test, we disinfected the chamber with 75% alcohol between every two experiments.

The elevated plus maze test and the open field test were conducted in a room maintained at 26°C. The lights were turned off, with only wall lights left on. The animals were placed in a dimly lit room for at least half an hour for adaptation. Each mouse was given 30 seconds of adaption after being put into the apparatus, followed by a five-minute formal recording.

### Immunofluorescence staining

Mice were anesthetized with intraperitoneal injection of pentobarbital sodium. Subsequently, they were perfused with 0.9% saline, followed by 4% paraformaldehyde. The brains were extracted and immersed in 4% paraformaldehyde at 4°C overnight for fixation, then dehydrated using sucrose solutions of 15% (w/v) and 30% (w/v) until they sank. Coronal brain sections (30 μm thick) were obtained using a cryogenic cryotome (Thermo Fisher Scientific, NX50, USA). For immunofluorescence analysis, the sections were first warmed at 37°C for one hour and then washed six times with Tris-Buffered Saline with Tween-20 (TBST, 20 mM Tris-HCl, 150 mM NaCl, 0.1% Tween 20, PH 7.3-7.4) for 10 minutes each, with agitation on a shaker. The sections were then blocked with a solution containing 10% donkey serum at 37°C for one hour. Following blocking, the sections were incubated overnight at 4°C with primary antibodies, including rabbit anti-c-Fos antibody (1:500; ab190289; Abcam, USA). The next day, the slices were rewarmed again to 37°C for one hour and washing six times with TBST. They were then incubated with Alexa Fluor® 488-conjugated secondary antibody (donkey anti-rabbit) at 37 °C for one hour. Following this incubation, the sections were washed six times again with TSBT and finally stained with DAPI (ab104139; Abcam, USA). Imaging was performed using a digital pathology biopsy scanner.

### Statistical

Statistical analyses were performed using GraphPad Prism 9. All experimental data were expressed as means ± standard errors of the mean (SEM). The PWTs between different groups were tested with two-way repeated-measures Analysis of Variance (ANOVA) with Tukey's post-hoc test. The normality of the distribution of continuous variables was assessed using the Shapiro–Wilk normality test. For normally distributed data, Student's t-test (two-tailed) and one-way ANOVA followed by Tukey's post-hoc test were used to compare means of two and multiple groups, respectively. Mixed repeated ANOVA was used to compare the expression of c-Fos across different bregma distance. A significance level of $p < 0.05$ was considered statistically significant.

# Results

## SNI mice exhibit pain hypersensitivity and anxiety-like behaviors

The SNI was employed to simulate neuropathic pain and assess the alterations in PWTs and emotional responses pre- and post-modeling (Fig 1A). Compared to the sham group, SNI mice exhibited a significant decrease in PWTs (Fig 1B). In EPMT, SNI mice spent less time exploring the open arm (Fig 1C and 1F). Similarly, SNI mice showed avoidance behavior towards the center zone in OFT (Fig 1D and 1G). However, their total distance traveled remained unchanged, indicating that their locomotor activity was unaffected following SNI surgery (Fig 1E). The results demonstrated that SNI mice exhibit pain hypersensitivity and anxiety-like behaviors.

## Expression of c-Fos in the BLA and CeA was upregulated in SNI mice

The amygdala has been identified as a crucial region implicated in the comorbidity of chronic neuropathic pain and affective disorders. Therefore, we investigated the c-Fos expression in both BLA and CeA, respectively (Fig 2A and 2C). The results revealed significantly higher c-Fos expression levels in both BLA and CeA when compared to the sham group,

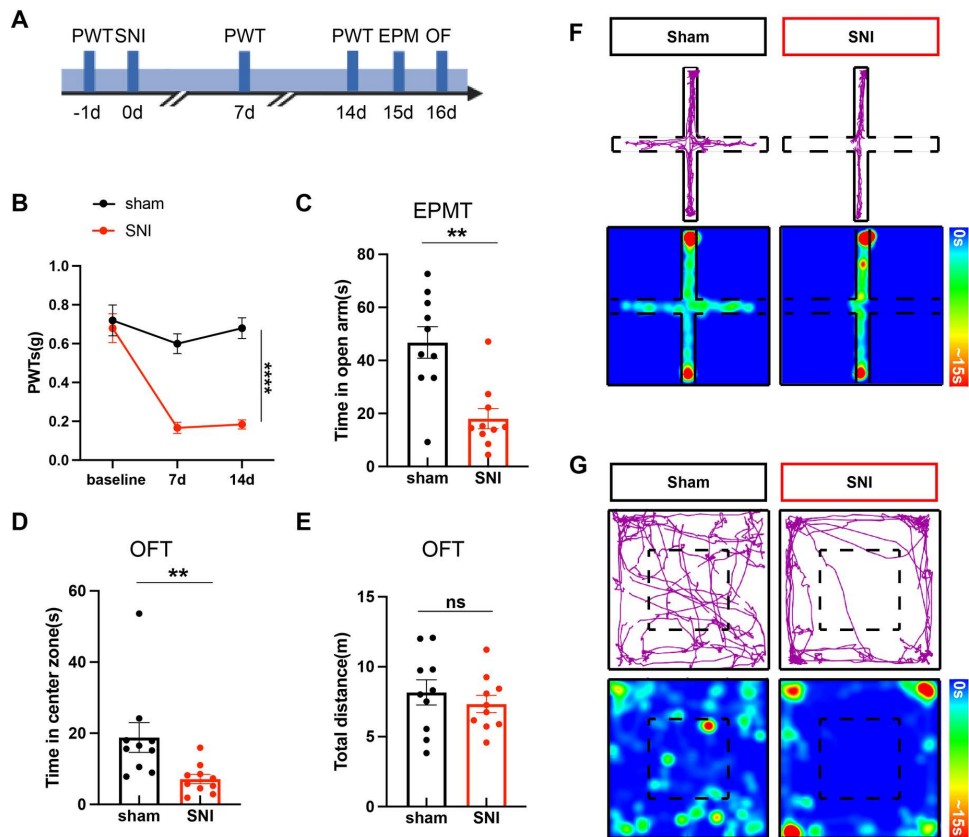

**Fig 1. SNI mice exhibit pain hypersensitivity and anxiety-like behaviors.** (A) Experimental timeline. (B) The PWTs of SNI mice and sham group (n = 10, $p < 0.0001$. Two-way ANOVA and Tukey's test). (C) Statistics of time in open arm in EPMT (n = 10, t = -4.076, $p = 0.001$. Student's t test). (D) Statistics of time in center zone in OFT (n = 10, t = -2.681, $p = 0.015$. Student's t test). (E) Statistics of total distance in OFT (n = 10, t = 0.757, $p = 0.459$. Student's t test). (F) Representative tracks and heat maps in EPMT. (G) Representative tracks and heat maps in OFT. Data presented as mean ± SEM, n = 10 mice per group. **$p < 0.01$, ****$p < 0.0001$ compared with sham group, ns, not significant.

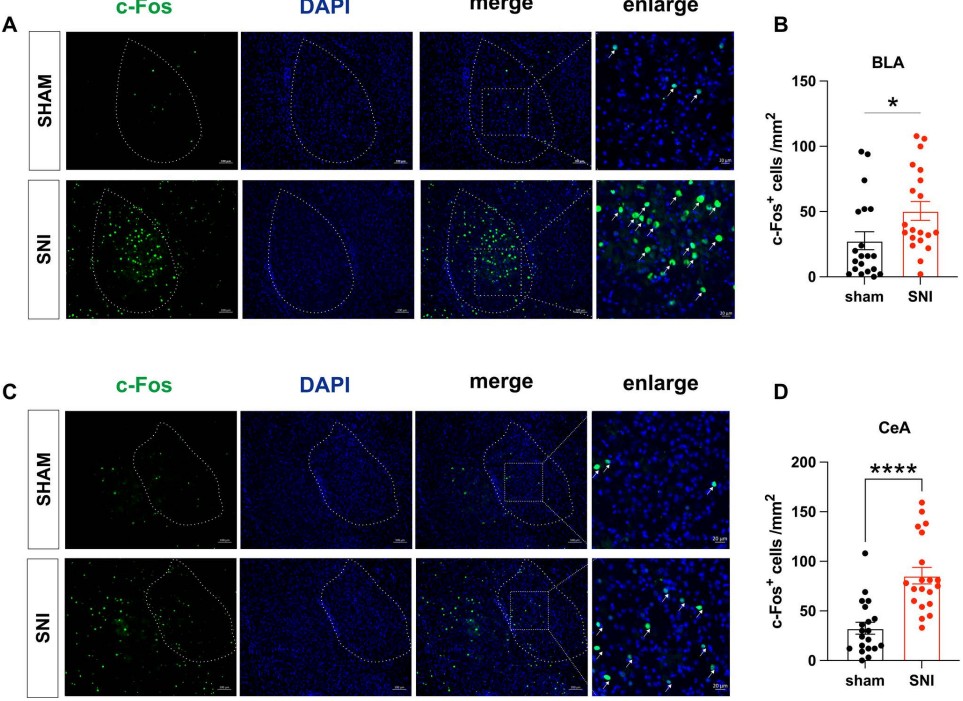

**Fig 2. Expression of c-Fos in the BLA and CeA was upregulated in SNI mice.** (A, C) Representative images of c-Fos in BLA and CeA. (B) The density of c-Fos expressed in BLA (n = 20, t = 2.292, $p$ = 0.028. Student's t test). (D) The density of c-Fos expressed in CeA (n = 20, t = 5.150, $p$ < 0.0001. Student's t test). Data are expressed as the mean ± SEM. $^*p$ < 0.05, $^{****}p$ < 0.0001 compared with sham mice.

following SNI surgery (Fig 2B and 2D). These findings demonstrate an increased activation of c-Fos in BLA and CeA subsequent to SNI surgery.

### The spatial variability of c-Fos expression in the BLA and CeA of SNI mice

Due to the substantial size of the amygdala, the BLA and CeA sections display distinct structural variations rostrocaudally. To further investigate their structure and function, we examined c-Fos expression in BLA and CeA of SNI mice and sham mice at different bregma distances (Fig 3A) and compared their respective density. We found a difference in c-Fos expression between SNI and sham beginning after 0.77mm in BLA and CeA. Besides, our findings indicated that as distance from bregma increases, there were spatial differences in c-Fos expression between BLA and CeA in male mice following SNI (Fig 3B and 3C).

### Discussion

Neuropathic pain, a prevalent form of chronic pain resulting from damage or disease of the somatosensory nervous system, is often accompanied by symptoms of anxiety, depression, and other emotional disturbances [26]. The amygdala, a crucial nucleus in the brain responsible for processing negative emotions [27], has been shown to play a role in anxiety regulation, as evidenced by its inactivation preventing anxiety [28]. The SNI model is a well-established and effective experimental model for studying neuropathic pain [29]. Following SNI induction in mice, hyperalgesia and anxiety-like behaviors have been observed [30,31], which were consistent with our own experimental findings. The present study provides compelling evidence for the role of BLA and CeA in mediating pain and anxiety symptoms in the SNI model of neuropathic pain. Our findings reveal a significant activation of these amygdala nuclei, as evidenced by increased c-Fos

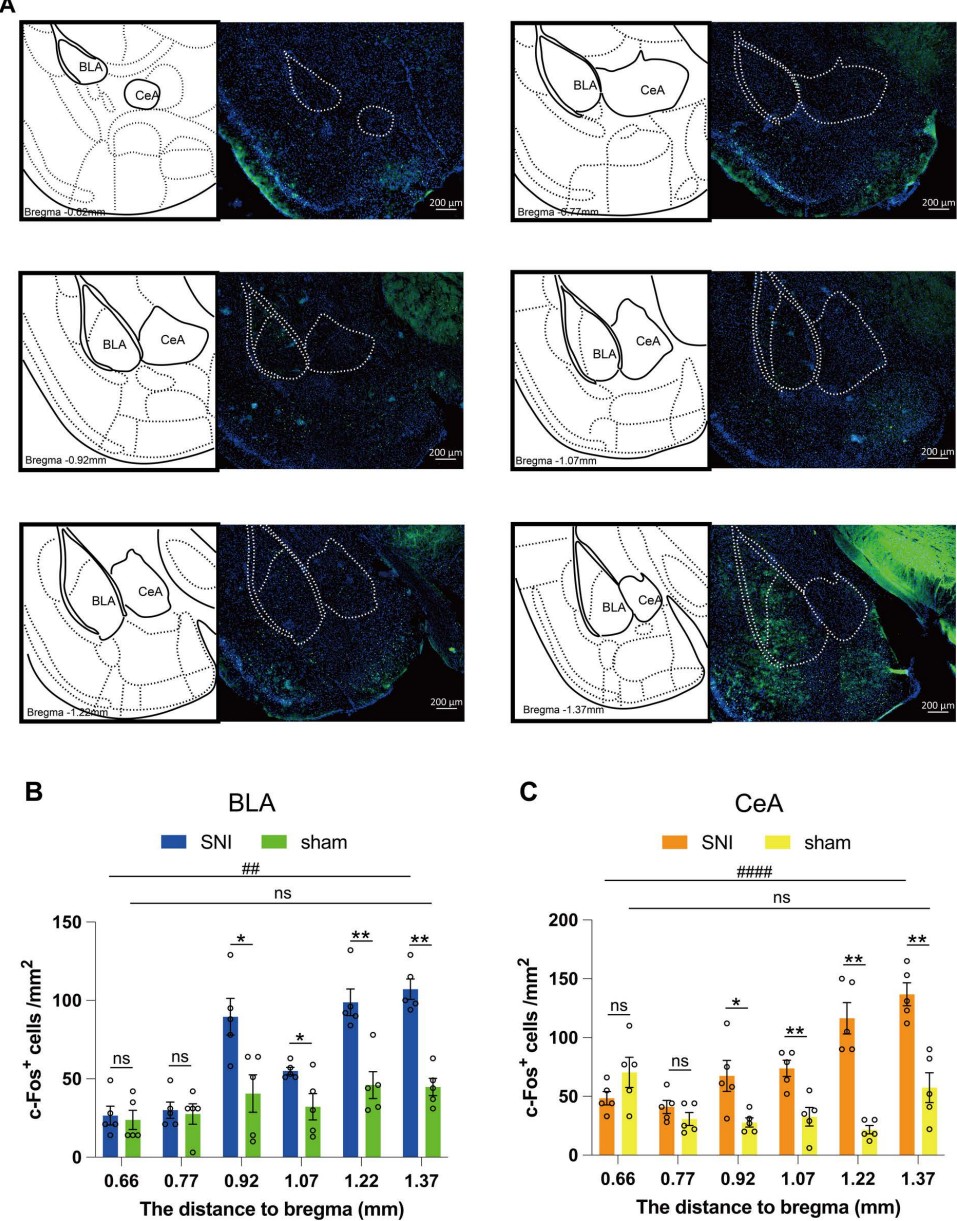

**Fig 3. The spatial variability of c-Fos expression in the BLA and CeA of SNI mice.** (A) Representative images of c-Fos (green) of BLA and CeA at different distance to bregma. (B) The c-Fos expression along the rostro-caudal BLA following SNI (n = 5). (C) The c-Fos expression along the rostro-caudal CeA following SNI (n = 5). Data are expressed as the mean ± SEM. $^*p < 0.05$, $^{**}p < 0.01$ compared between SNI and sham, ns, not significance. $^#p < 0.05$, $^{####}p < 0.0001$ compared the extent of c-Fos expression along the rostro-caudal the BLA and CeA, ns, not significance. (mixed repeated ANOVA).

expression, in response to peripheral nerve injury. This observation aligns with previous research highlighting the involvement of the amygdala in the processing of nociceptive and emotional information. Chronic neuropathic pain in mice has been shown to significantly impact the structure and function of various brain regions [32], potentially contributing to the development of anxiety-like behaviors.

One of the novel aspects of our study is the investigation of the spatial distribution of c-Fos expression within the amygdala, relative to the bregma landmark. We discovered that there were spatial differences in c-Fos expression in the

BLA and CeA of SNI mice rostrocaudally. This spatial gradient of activation within the amygdala may suggest a graded response to peripheral nociceptive input. Such a finding adds a new dimension to our understanding of amygdala processing in pain and anxiety and suggests that future studies should consider the potential significance of topographical differences within this complex structure.

It is worth mentioning that CeA has been reported in the literature to produce an analgesic effect when in the activated state, which seems to be contrary to our findings [33,34]. We believe that this is related to the types of neurons contained in the BLA and CeA. Most of the BLA is composed of excitatory neurons, while CeA is mainly composed of inhibitory neurons. There must be some circuit connection between them, and they cooperate to form the amygdala processing pain and pain-related anxiety. This is also our next research content.

The activation of the BLA and CeA in the context of neuropathic pain is particularly intriguing given their well-established roles in fear learning and emotional memory. Our results suggest that, in addition to their classic roles in emotional processing, these amygdala nuclei may also contribute to the emotional aspects of pain, such as anxiety and fear avoidance, which are frequently comorbid with chronic pain conditions. By identifying a specific anatomical correlate of this interaction, our study opens up new avenues for targeted therapeutic interventions aimed at mitigating the emotional burden associated with neuropathic pain.

It should be noted that a major limitation of this study is that we used only male mice as experimental subjects. Gender differences play an important role in pain perception, anxiety response and nervous system function. Therefore, our results may not be fully representative of the responses of female mice under the same conditions. In particular, given the potential influence of sex hormones on pain processing and emotion regulation, behavioral changes and neural mechanisms in female mice following nerve injury may differ from those in male mice. A research report has indicated that female mice exhibit more anxiety-related behaviors, and they also demonstrate heightened sensitivity to thermal pain [35]. It has also been shown that there are differences in protein degradation between female and male rats during fear memory formation in the amygdala [36]. To more fully understand the performance of nerve injury-induced hyperalgesia and anxiety-related behaviors in different genders, future studies should consider including female mice and explore the specific effects of sex differences on study outcomes.

However, it is important to acknowledge that the c-Fos technique, while informative, is a relatively indirect measure of neuronal activation. Future studies utilizing more direct measures of neuronal activity, such as optogenetics or electrophysiology, will be essential for elucidating the precise mechanisms underlying amygdala involvement in pain and anxiety. Furthermore, given the intricate interconnectivity of the amygdala with other brain regions implicated in pain and emotion processing (e.g., the prelimbic cortex, anterior cingulate cortex, and paraventricular thalamus), further investigations into the functional circuitry linking these structures may yield important insights into the etiology and treatment of neuropathic pain and its comorbidities.

## Conclusion

In conclusion, our study highlights the crucial role of the BLA and CeA in mediating pain and anxiety symptoms in the SNI model of neuropathic pain. The spatial gradient of c-Fos expression within the amygdala, as demonstrated in our findings, provides a novel perspective on amygdala processing in the context of peripheral nerve injury. These results have important implications for our understanding of the neural underpinnings of chronic pain and its emotional consequences and may ultimately contribute to the development of more effective treatments for this devastating condition.

## Supporting information

**S1 Data. Data of Fig 1.**
(XLSX)

**S2 Data. Data of Fig 2.**
(XLSX)

**S3 Data. Data of Fig 3.**
(XLSX)

## Author contributions

**Conceptualization:** Siyuan Tong, Yuerong Chen.

**Data curation:** Siyuan Tong, Yuerong Chen.

**Formal analysis:** Siyuan Tong, Yuerong Chen.

**Funding acquisition:** Yuanyuan Wu.

**Investigation:** Zonglin Wu, Yuxin Wei, Yuanyuan Wu.

**Methodology:** Siyuan Tong, Zonglin Wu, Yuxin Wei, Xirui Wang, Yuanyuan Wu.

**Project administration:** Yuxin Wei, Xirui Wang, Yuanyuan Wu.

**Resources:** Zonglin Wu, Linbao Wang, Xirui Wang.

**Software:** Siyuan Tong, Yuerong Chen, Linbao Wang, Xirui Wang.

**Supervision:** Yuerong Chen.

**Validation:** Linbao Wang.

**Visualization:** Linbao Wang.

**Writing – original draft:** Siyuan Tong.

**Writing – review & editing:** Siyuan Tong.

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
