## [Decision Letter · Decision Letter 0]

26 Aug 2024

PONE-D-24-27569Nerve injury induces pain hypersensitivity and anxiety-related behaviors via amygdala activation in male micePLOS ONE

Dear Dr. Wu,

Thank you for submitting your manuscript to PLOS ONE. After careful consideration, we feel that it has merit but does not fully meet PLOS ONE’s publication criteria as it currently stands. Therefore, we invite you to submit a revised version of the manuscript that addresses the points raised during the review process.

We look forward to receiving your revised manuscript.

Kind regards,

John M. Streicher, Ph.D.

Academic Editor

PLOS ONE

Journal Requirements:

"The National Natural Science Foundation of China (NO. 8207151734), the Natural Science Foundation of Zhejiang Province (NO. LY23H270009)"

4. PLOS requires an ORCID iD for the corresponding author in Editorial Manager on papers submitted after December 6th, 2016. Please ensure that you have an ORCID iD and that it is validated in Editorial Manager. To do this, go to ‘Update my Information’ (in the upper left-hand corner of the main menu), and click on the Fetch/Validate link next to the ORCID field. This will take you to the ORCID site and allow you to create a new iD or authenticate a pre-existing iD in Editorial Manager. Please see the following video for instructions on linking an ORCID iD to your Editorial Manager account: https://www.youtube.com/watch?v=_xcclfuvtxQ"

Additional Editor Comments:

Dear Authors,

The concerns raised by the Reviewers are significant, and will require extensive revision to address. The point raised by Reviewer 2 that these experiments have been reported numerous times in the literature has merit; however, the policy of PLOS ONE is to publish only based on scientific quality, not impact or novelty. These concerns thus do not preclude publication, however, the history of previous experimentation should be addressed in the Introduction and Discussion and any relevant differences highlighted.

Reviewers' comments:

Reviewer's Responses to Questions

**Comments to the Author**

1. Is the manuscript technically sound, and do the data support the conclusions?

Reviewer #1: Partly

Reviewer #2: Partly

2. Has the statistical analysis been performed appropriately and rigorously? 

Reviewer #1: Yes

Reviewer #2: Yes

3. Have the authors made all data underlying the findings in their manuscript fully available?

Reviewer #1: No

Reviewer #2: Yes

4. Is the manuscript presented in an intelligible fashion and written in standard English?

Reviewer #1: No

Reviewer #2: Yes

5. Review Comments to the Author

Reviewer #1: The manuscript by Tong and colleagues provides valuable insights into neuropathic pain-induced anxiety-like behavior in rodents, offering a potential animal model to study the neural basis of pain and the comorbid negative affective states. However, the manuscript lacks clarity, and additional experiments would be beneficial to enhance the impact of the research. Below are my general suggestions for improving the quality and novelty of the manuscript to be considered for publication in PLOS ONE:

1. English Language and Clarity: The manuscript’s use of English needs significant improvement. The writing is often difficult to follow, and, in some instances, it presents conflicting ideas within the same paragraph. For example, in lines 132 and 133, it is stated that SNI both changes and remains unaffected in terms of distance traveled within the same sentence.

2. Inclusion of Female Mice: The study would be more impactful and generate greater interest if female mice were also included. Previous literature (e.g., Burek, 2022; Lorente et al., 2024) has demonstrated a sex-dependent effect of pain on anxiety-like behaviors. In fact, male mice sometimes fail to exhibit increased anxiety-like behavior in the presence of pain.

3. Introduction and Literature Context: The introduction should be better grounded in existing literature. The selection of the amygdala as the nucleus of interest is relevant to the field, but providing more detailed data from previously published studies would strengthen the rationale for focusing on the amygdala in this context.

4. Timing of Testing: The timing of testing is not specified, which is crucial as rodent behavior can be influenced by the time of day in relation to the light cycle. Please specify the testing times, considering the light cycle.

5. TBST Composition: More details on the composition of TBST should be provided, particularly concerning the concentration of the detergent used.

6. Discussion of Contradictory Data: The discussion should address contradictory findings that have been published in the existing literature.

7. Analysis of PrL, PVT, and PAG: While the roles of the PrL, PVT, and PAG are of significant interest, it is difficult to draw conclusions about their involvement without analyzing these areas. If the researchers still have access to the brains of the animals used, including an analysis of these regions would be fundamental to substantiate claims regarding their roles.

8. Minor Comments: Abbreviations should be used consistently throughout the manuscript, and the English should be further refined.

Reviewer #2: In this manuscript Wu et al, assess neuropathic pain impact on hyperalgesia and anxiety-like behaviors using VonFrey and Elevated Plus Maze/Open Field Tests, respectively.

The authors provide an extremely brief introduction lacking proper context or rigor to present their investigations. This is followed by a brief presentation of the methods used, a very succinct description of the results, and a discussion in which many elements are disconnected from the data shown in the manuscript.

Overall, :

1. the experiments shown by the authors have been repeated on multiple occasions in the literature

2. there is a lack of description for the Sham control group - did the controls received a Sham surgery or no procedure at all

3. the CFos experiment is not associated with specific acute painful event, which might be fair but at least needs to be discussed

4. cFos analysis is presented as number of Fos positive neurons. the authors should consider a density of Fos positive per mm3 at least

5. the "positive correlation with the distance from Bregma" is nebulous and needs to be discussed in terms of rostro-caudal involvement of BLA/CeA in pain if needs to be presented.

Overall, this manuscript is extremely succinct, mostly descriptive, and repeats findings that have been done on many occasions in the past years.

6. PLOS authors have the option to publish the peer review history of their article (what does this mean? ). If published, this will include your full peer review and any attached files.

**Do you want your identity to be public for this peer review?** For information about this choice, including consent withdrawal, please see our Privacy Policy .

Reviewer #1: No

Reviewer #2: No

---

## [Author Response · Author response to Decision Letter 0]

13 Oct 2024

Reviewer #1: The manuscript by Tong and colleagues provides valuable insights into neuropathic pain-induced anxiety-like behavior in rodents, offering a potential animal model to study the neural basis of pain and the comorbid negative affective states. However, the manuscript lacks clarity, and additional experiments would be beneficial to enhance the impact of the research. Below are my general suggestions for improving the quality and novelty of the manuscript to be considered for publication in PLOS ONE:

1. English Language and Clarity: The manuscript’s use of English needs significant improvement. The writing is often difficult to follow, and, in some instances, it presents conflicting ideas within the same paragraph. For example, in lines 132 and 133, it is stated that SNI both changes and remains unaffected in terms of distance traveled within the same sentence.

Response:

We thank you for the suggestion. We regret the mistake and have corrected the inaccurate description (page 5, line 158). In addition, we further checked the language of the article to make the expression more accurate.

2. Inclusion of Female Mice: The study would be more impactful and generate greater interest if female mice were also included. Previous literature (e.g., Burek, 2022; Lorente et al., 2024) has demonstrated a sex-dependent effect of pain on anxiety-like behaviors. In fact, male mice sometimes fail to exhibit increased anxiety-like behavior in the presence of pain.

Response:

Thanks for your valuable suggestion. Before we started this experiment, we did have the stereotype that male mice might be more stable in behavior. We also followed the previous research paradigm involving only male mice [1-3]. In the follow-up experiments, we will also use female mice as our research subjects to further investigate this sex-dependent effect. Thanks again for your valuable comments.

[1] Xu Y, Zhu X, Chen Y, et al. Electroacupuncture alleviates mechanical allodynia and anxiety-like behaviors induced by chronic neuropathic pain via regulating rostral anterior cingulate cortex-dorsal raphe nucleus neural circuit. CNS Neurosci Ther. 2023;29(12):4043-4058. doi:10.1111/cns.14328

[2] Chen Y, Tong S, Xu Y, et al. Involvement of basolateral amygdala-rostral anterior cingulate cortex in mechanical allodynia and anxiety-like behaviors and potential mechanisms of electroacupuncture. CNS Neurosci Ther. 2024;30(9):e70035. doi:10.1111/cns.70035

[3] Wu Y, Chen Y, Xu Y, et al. Proteomic Analysis of the Amygdala Reveals Dynamic Changes in Glutamate Transporter-1 During Progression of Complete Freund's Adjuvant-Induced Pain Aversion. Mol Neurobiol. 2023;60(12):7166-7184. doi:10.1007/s12035-023-03415-7

3. Introduction and Literature Context: The introduction should be better grounded in existing literature. The selection of the amygdala as the nucleus of interest is relevant to the field, but providing more detailed data from previously published studies would strengthen the rationale for focusing on the amygdala in this context.

Response

Thanks for your valuable advice. We admit that the introduction in our previous manuscript is too simple. In response to your suggestion, we have revised the introduction to include more detailed data and information from previously published studies. We have also included additional references to relevant studies. These additions aim to strengthen the rationale for our focus on the amygdala and to provide a clearer context for our study within the existing body of literature. (page 2-3, line 32-78).

4. Timing of Testing: The timing of testing is not specified, which is crucial as rodent behavior can be influenced by the time of day in relation to the light cycle. Please specify the testing times, considering the light cycle.

Response

Thanks for your advice. We would like to clarify that all testing was conducted during the light phase of the animals' diurnal cycle, specifically between 9 AM and 3 PM. This time frame was chosen to minimize any potential effects of the light cycle on the behavioral outcomes. We have revised and supplemented the first draft accordingly to more accurately describe our experimental methods (page 4, line128-129).

5. TBST Composition: More details on the composition of TBST should be provided, particularly concerning the concentration of the detergent used.

Response:

We apologize for not including this information in the initial submission and have now updated our methods section to include the detailed composition of TBST (page 5, line 137-138). The revised details are as follows:

TBST buffer composition:

Tris-buffered saline (TBS): 20 mM Tris-HCl, 150 mM NaCl, pH 7.4

Tween 20: 0.1% (v/v)

Thank you once again for your valuable feedback.

6. Discussion of Contradictory Data: The discussion should address contradictory findings that have been published in the existing literature.

Response:

Thank you for your suggestion. In response to your comment, we have revised the discussion section of our manuscript to include an analysis of the contradictory data published in the literature. We have carefully reviewed relevant studies and provided our interpretation of how these findings relate to our own results. We have revised the discussion section to address and discuss any contradictory findings reported in the existing literature, ensuring a comprehensive analysis of the topic (page 6, line 199-204). Thank you once again for your valuable feedback and suggestions.

7. Analysis of PrL, PVT, and PAG: While the roles of the PrL, PVT, and PAG are of significant interest, it is difficult to draw conclusions about their involvement without analyzing these areas. If the researchers still have access to the brains of the animals used, including an analysis of these regions would be fundamental to substantiate claims regarding their roles.

Response:

Thank you for highlighting the importance of analyzing the PrL, PVT, and PAG regions. We acknowledge that our current data do not directly address the involvement of these areas. Unfortunately, we no longer have access to the brains of the animals used in this study. However, we plan to include an analysis of these regions in our future studies to further investigate their roles and substantiate our claims.

8. Minor Comments: Abbreviations should be used consistently throughout the manuscript, and the English should be further refined.

Response:

Thanks for the suggestion. We have reviewed the manuscript carefully and ensured that all abbreviations are used consistently throughout the text. As for the refinement of the English language, we understand that clear and concise writing is essential for effective communication of our research findings. We have taken your comments into consideration and have revised the manuscript to improve the clarity and precision of our language. We have also proofread the text carefully to eliminate any grammatical errors or awkward phrasing. We are grateful for your suggestions and believe that they have helped us to improve the quality of our manuscript.

Reviewer #2: In this manuscript Wu et al, assess neuropathic pain impact on hyperalgesia and anxiety-like behaviors using VonFrey and Elevated Plus Maze/Open Field Tests, respectively.

The authors provide an extremely brief introduction lacking proper context or rigor to present their investigations. This is followed by a brief presentation of the methods used, a very succinct description of the results, and a discussion in which many elements are disconnected from the data shown in the manuscript.

Overall:

1. the experiments shown by the authors have been repeated on multiple occasions in the literature

Response:

Thank you for your valuable advice. We acknowledge the significance of these previous studies and their contributions to the field. Our intention in including these experiments was not to claim novelty, but rather to provide a robust foundation for our subsequent analyses and conclusions. By repeating the experiments, we aimed to confirm the consistency and reproducibility of the results, which strengthens the overall confidence in the findings. We believe that our study adds value by providing a fresh perspective and contributing to the ongoing dialogue in the field. And this is also the basis of our next research project. We appreciate your valuable feedback and have made the necessary adjustments to our manuscript.

2. there is a lack of description for the Sham control group - did the controls received a Sham surgery or no procedure at all

Response:

Thank you for raising the concern regarding the description of the Sham control group. We apologize for any lack of clarity in our manuscript. For the sham group, mice underwent the same procedure but without nerve ligation or transection. We have now added a more detailed description of the Sham surgery in the Methods section to ensure transparency and reproducibility of our results (page 4, line 95-96).

3. the CFos experiment is not associated with specific acute painful event, which might be fair but at least needs to be discussed

Response

Thank you very much for your valuable feedback. We acknowledge that our study focuses on chronic pain and, as such, the design does not directly link c-Fos expression to a particular acute painful stimulus. However, we believe that this approach is justified in the context of chronic pain research, as it allows us to investigate neuronal activation patterns associated with sustained pain states rather than transient, acute events [1]. Our future studies will utiliz more direct measures of neuronal activity, such as calcium imaging or electrophysiology, will be essential for elucidating the precise mechanisms underlying amygdala involvement in pain and anxiety. We also added a discussion of the limitations of C-Fos experiments (page 6, line 212-215).

[1] Xu Y, Zhu X, Chen Y, et al. Electroacupuncture alleviates mechanical allodynia and anxiety-like behaviors induced by chronic neuropathic pain via regulating rostral anterior cingulate cortex-dorsal raphe nucleus neural circuit. CNS Neurosci Ther. 2023;29(12):4043-4058. doi:10.1111/cns.14328

4. cFos analysis is presented as number of Fos positive neurons. the authors should consider a density of Fos positive per mm3 at least

Response:

Thank you for your suggestion. We appreciate your point that presenting the data as the number of Fos-positive neurons alone may not fully capture the underlying biology. To address this, we recalculated the density of c-Fos per square millimeter in the BLA and CeA across different sections (Fig 3B).

5. the "positive correlation with the distance from Bregma" is nebulous and needs to be discussed in terms of rostro-caudal involvement of BLA/CeA in pain if needs to be presented.

Response:

Thank you for the valuable suggestion. We understand that the presentation may have been nebulous. We have clarified the anatomical significance of Bregma as a reference point in neuroscientific research and explained how the distance from Bregma can serve as a proxy for the rostro-caudal axis within the brain. We have delved deeper into the rostro-caudal distribution of BLA and CeA, highlighting the regional differences in their involvement in pain modulation. To address this, we have revised the manuscript to include a discussion on the rostro-caudal involvement of the BLA and CeA in pain regulation (page 5-6, line171-175). We believe that these revisions have significantly improved the clarity and depth of our discussion, and we are grateful for your guidance in helping us refine our manuscript.

---

## [Decision Letter · Decision Letter 1]

2 Dec 2024

PONE-D-24-27569R1Nerve injury induces pain hypersensitivity and anxiety-related behaviors via amygdala activation in male micePLOS ONE

Dear Dr. Wu,

Thank you for submitting your manuscript to PLOS ONE. After careful consideration, we feel that it has merit but does not fully meet PLOS ONE’s publication criteria as it currently stands. Therefore, we invite you to submit a revised version of the manuscript that addresses the points raised during the review process.

We look forward to receiving your revised manuscript.

Kind regards,

John M. Streicher, Ph.D.

Academic Editor

PLOS ONE

Additional Editor Comments:

I appreciate your efforts at revision. However, both Reviewers still have significant concerns, including to the basic scientific rigor of the work. I will thus allow one more attempt at Major Revision to satisfy these concerns.

Reviewers' comments:

Reviewer's Responses to Questions

**Comments to the Author**

1. If the authors have adequately addressed your comments raised in a previous round of review and you feel that this manuscript is now acceptable for publication, you may indicate that here to bypass the “Comments to the Author” section, enter your conflict of interest statement in the “Confidential to Editor” section, and submit your "Accept" recommendation.

Reviewer #1: (No Response)

Reviewer #2: All comments have been addressed

2. Is the manuscript technically sound, and do the data support the conclusions?

Reviewer #1: Partly

Reviewer #2: Partly

3. Has the statistical analysis been performed appropriately and rigorously? 

Reviewer #1: No

Reviewer #2: Yes

4. Have the authors made all data underlying the findings in their manuscript fully available?

Reviewer #1: No

Reviewer #2: Yes

5. Is the manuscript presented in an intelligible fashion and written in standard English?

Reviewer #1: No

Reviewer #2: Yes

6. Review Comments to the Author

Reviewer #1: Although the authors have made improvements to the manuscript, some concerns remain only partially addressed. I believe these issues need to be fully resolved to ensure the manuscript's quality. Additionally, a more in-depth analysis of existing data and a refined discussion are essential.

Major Concerns:

1. English Language: the manuscript’s use of English still requires further improvement, particularly in terms of readability and clarity.

2. Inclusion of Female Mice: if including female mice is not feasible, the authors must provide a more thorough discussion of sex differences by referencing data from other studies. They should also explicitly acknowledge that the exclusion of female mice is a limitation of this study. The justification provided—reliance on previous studies—no longer holds, as studies now often include female mice.

3. Testing Time and Light Cycle Effects: while the authors state that testing occurred between 9 a.m. and 3 p.m. to avoid light cycle effects, they have not provided data to substantiate this claim. To my knowledge, there are no studies proving that testing within this range minimizes light cycle effects. Importantly, behavioral testing conducted during the light phase (inactive phase) is less optimal, as it may yield different conclusions compared to testing during the active (dark) phase. This should be addressed in the discussion and compared with data from other studies. Furthermore, the exact timing of the light/dark cycle (lights on/off) should be reported.

4. Discussion: while the discussion has improved, further refinement is needed. It should include a detailed examination of sex differences, the timing of testing, and a deeper comparison with existing literature.

5. Statistical Analysis: the explanation of statistical methods is insufficient. Although a t-test may be acceptable for behavioral analysis with two groups, the manuscript does not indicate whether normality of the data was tested. Additionally, t-values are missing from the results section. For spatial analysis, a t-test is inappropriate. An ANOVA is necessary to compare conditions (SNI vs. sham) across different bregma distances.

Minor Comments:

1. c-Fos Assumptions:

c-Fos is an immediate early gene whose expression increases shortly after neuronal activation. It is induced through calcium-dependent signaling pathways in response to excitatory input. While c-Fos levels may rise due to neuronal activation associated with injury or pain, it is not strictly indicative of nociception. The manuscript should avoid making this assumption.

Reviewer #2: I would like to thank the authors for taking into account my comments on their previous submission.

While the authors have answered some of my comments there are still some details left unresolved

1) The figure 2 still shows number of CFos neurons while the authors have shown these results as CFos+ cells/mm2. The comment previously made applied to all data using CFos

2) while there is a very brief discussion about the rostro-caudal axis data presented in Figure 3, it would be a good idea to compare the CFOs+ density in SNI versus Sham animals. This would provide a better picture of the pattern of activity in CeA/BLA induced by pain. Assuming of course that this dat comes from the same tissue as the one presented in Fig2 this should not be a complex analysis to run/add to the manuscript.

7. PLOS authors have the option to publish the peer review history of their article (what does this mean? ). If published, this will include your full peer review and any attached files.

**Do you want your identity to be public for this peer review?** For information about this choice, including consent withdrawal, please see our Privacy Policy .

Reviewer #1: No

Reviewer #2: No

---

## [Author Response · Author response to Decision Letter 1]

5 Jan 2025

Reviewer #1: Although the authors have made improvements to the manuscript, some concerns remain only partially addressed. I believe these issues need to be fully resolved to ensure the manuscript's quality. Additionally, a more in-depth analysis of existing data and a refined discussion are essential.

Major Concerns:

1. English Language: the manuscript’s use of English still requires further improvement, particularly in terms of readability and clarity.

Response:

Thank you for your feedback on our manuscript. We acknowledge that the readability and clarity of the English could be improved. We thoroughly reviewed the manuscript to ensure that each point was conveyed in a logical and concise manner. This will help to improve the overall readability of the text.

2. Inclusion of Female Mice: if including female mice is not feasible, the authors must provide a more thorough discussion of sex differences by referencing data from other studies. They should also explicitly acknowledge that the exclusion of female mice is a limitation of this study. The justification provided—reliance on previous studies—no longer holds, as studies now often include female mice.

Response:

Thank you for your constructive suggestion. We fully acknowledge that the exclusion of female mice is a limitation of our current work. In response to your comments, we revised the manuscript to include a more comprehensive discussion of sex differences, referencing relevant data from recent studies that have included female mice. We understand that the reliance on previous studies, which may have excluded female mice, is no longer sufficient justification for our current approach. We acknowledge this limitation in the revised manuscript and emphasize the importance of future studies to address this gap in the literature. Thank you again for your valuable feedback. (page 6-7, line 218-229)

3. Testing Time and Light Cycle Effects: while the authors state that testing occurred between 9 a.m. and 3 p.m. to avoid light cycle effects, they have not provided data to substantiate this claim. To my knowledge, there are no studies proving that testing within this range minimizes light cycle effects. Importantly, behavioral testing conducted during the light phase (inactive phase) is less optimal, as it may yield different conclusions compared to testing during the active (dark) phase. This should be addressed in the discussion and compared with data from other studies. Furthermore, the exact timing of the light/dark cycle (lights on/off) should be reported.

Response

Thank you for bringing to our attention the concerns regarding the testing time and potential light cycle effects on our study results. We would like to clarify that our behavioral testing was conducted in a dark room with wall lights, which was specifically designed to minimize any external light influences. We are sorry that we did not make this explicit in our previous manuscript. The testing room was equipped with blackout curtains and light-tight doors to ensure that no ambient light could penetrate during the experiments. Therefore, despite the fact that testing occurred between 9 a.m. and 3 p.m., the animals were exposed to consistent dimly lit conditions throughout the testing period. We understand that the timing of behavioral tests relative to the animals' light/dark cycle is an important consideration. However, in our study, the dark room conditions allowed us to create a controlled environment that was independent of the external light cycle. This allowed us to perform the tests at a consistent time each day without concern for light cycle effects. To address this point, we made the corresponding supplement in the method on the specific conditions of our testing environment. Thank you again for your thoughtful comments and suggestions. (page 4, line 126-129)

4. Discussion: while the discussion has improved, further refinement is needed. It should include a detailed examination of sex differences, the timing of testing, and a deeper comparison with existing literature.

Response

Thank you for your constructive feedback on our manuscript's discussion section. Regarding the need for a detailed examination of sex differences, we acknowledge that this is an important consideration in our study. We referenced relevant data from recent literature that have included female mice. In the revised discussion, we will provide a more thorough analysis of any observed sex-specific effects and discuss their potential implications for our findings. (page 6-7, line 218-229)

Concerning the timing of testing, we understand that this can be a critical factor in behavioral studies. Our tests were conducted in a controlled environment (as previously mentioned, in a dark room to minimize light cycle effects). It is worth mentioning that our laboratory has developed relatively mature testing protocols for both the elevated plus maze and open field tests, and the current experiment follows the traditional operational paradigms established in our lab [1-5]. Taking these into consideration, we have added specific supplements in the Methods section of the manuscript and have refrained from elaborating excessively on these points in the Discussion section.

We made these refinements to the discussion section promptly and ensure that the revised manuscript provides a comprehensive and nuanced analysis of our results in the context of current research.

Thank you once again for your valuable input.

[1] Wang YJ, Zan GY, Xu C, et al. The claustrum-prelimbic cortex circuit through dynorphin/κ-opioid receptor signaling underlies depression-like behaviors associated with social stress etiology. Nat Commun. 2023;14(1):7903. Published 2023 Nov 30. doi:10.1038/s41467-023-43636-x

[2] Chen Y, Tong S, Xu Y, et al. Involvement of basolateral amygdala-rostral anterior cingulate cortex in mechanical allodynia and anxiety-like behaviors and potential mechanisms of electroacupuncture. CNS Neurosci Ther. 2024;30(9):e70035. doi:10.1111/cns.70035

[3] Xie Y, Shen Z, Zhu X, et al. Infralimbic-basolateral amygdala circuit associated with depression-like not anxiety-like behaviors induced by chronic neuropathic pain and the antidepressant effects of electroacupuncture. Brain Res Bull. 2024;218:111092. doi:10.1016/j.brainresbull.2024.111092

[4] Zhu X, Zhang C, Hu Y, et al. Modulation of Comorbid Chronic Neuropathic Pain and Anxiety-Like Behaviors by Glutamatergic Neurons in the Ventrolateral Periaqueductal Gray and the Analgesic and Anxiolytic Effects of Electroacupuncture. eNeuro. 2024;11(8):ENEURO.0454-23.2024. Published 2024 Aug 28. doi:10.1523/ENEURO.0454-23.2024

[5] Wu Z, Shen Z, Xu Y, et al. A neural circuit associated with anxiety-like behaviors induced by chronic inflammatory pain and the anxiolytic effects of electroacupuncture. CNS Neurosci Ther. 2024;30(4):e14520. doi:10.1111/cns.14520

5. Statistical Analysis: the explanation of statistical methods is insufficient. Although a t-test may be acceptable for behavioral analysis with two groups, the manuscript does not indicate whether normality of the data was tested. Additionally, t-values are missing from the results section. For spatial analysis, a t-test is inappropriate. An ANOVA is necessary to compare conditions (SNI vs. sham) across different bregma distances.

Response

Thank you for bringing these points to our attention. Regarding the normality testing, we acknowledge that we did not explicitly mention it in the manuscript. The normality of the distribution of continuous variables was assessed using the Shapiro–Wilk normality test. We added this information in the revised Methods section to ensure clarity. We also apologized for the oversight in not reporting the t-values in the Results section. We included these values in the revised manuscript to provide a complete picture of the statistical analysis. For the spatial analysis, we agree that a t-test may not be the most appropriate method for comparing conditions across different bregma distances. We conducted an ANOVA to analyze these data and reported the results in the revised manuscript. (page 5, line 148-156)

Minor Comments:

1. c-Fos Assumptions:

c-Fos is an immediate early gene whose expression increases shortly after neuronal activation. It is induced through calcium-dependent signaling pathways in response to excitatory input. While c-Fos levels may rise due to neuronal activation associated with injury or pain, it is not strictly indicative of nociception. The manuscript should avoid making this assumption.

Response

Thank you for highlighting the important distinction between c-Fos expression and nociception. We fully agree that c-Fos is an immediate early gene whose expression can increase in response to neuronal activation associated with various stimuli, including injury or pain, but it is not a direct indicator of nociception. We will revise the manuscript to clarify this point and avoid making any assumptions that link c-Fos expression directly to nociception. Instead, we will emphasize that c-Fos expression is a marker of neuronal activation and should be interpreted in the context of the experimental paradigm and other available data. Thank you for your valuable feedback, and we appreciate your time and effort in reviewing our manuscript. (page 2, line 18; page 3, line 75)

Reviewer #2: I would like to thank the authors for taking into account my comments on their previous submission.

While the authors have answered some of my comments there are still some details left unresolved

1) The figure 2 still shows number of CFos neurons while the authors have shown these results as CFos+ cells/mm2. The comment previously made applied to all data using CFos

Response:

Thank you for bringing this inconsistency to our attention. You are correct that Figure 2 displays the number of c-Fos neurons, whereas we have presented the results elsewhere in the manuscript as c-Fos+ cells per square millimeter (c-Fos+ cells/mm²). To address this issue, we will revise Figure 2 to ensure that the units are consistent with the rest of the data presented. As shown below

2) while there is a very brief discussion about the rostro-caudal axis data presented in Figure 3, it would be a good idea to compare the CFOs+ density in SNI versus Sham animals. This would provide a better picture of the pattern of activity in CeA/BLA induced by pain. Assuming of course that this dat comes from the same tissue as the one presented in Fig2 this should not be a complex analysis to run/add to the manuscript.

Response:

Thank you for your valuable suggestion. Comparing the cfos+ density between SNI and Sham animals in the CeA/BLA is indeed a promising idea. We have already conducted immunofluorescence staining statistics on the brain slices from the existing sham group. As shown below

Thank you once again for your valuable suggestions.

---

## [Decision Letter · Decision Letter 2]

30 Mar 2025

PONE-D-24-27569R2Nerve injury induces pain hypersensitivity and anxiety-related behaviors via amygdala activation in male micePLOS ONE

Dear Dr. Wu,

Thank you for submitting your manuscript to PLOS ONE. After careful consideration, we feel that it has merit but does not fully meet PLOS ONE’s publication criteria as it currently stands. Therefore, we invite you to submit a revised version of the manuscript that addresses the points raised during the review process.

We look forward to receiving your revised manuscript.

Kind regards,

Armando Almeida

Academic Editor

PLOS ONE

Journal Requirements:

Reviewers' comments:

Reviewer's Responses to Questions

**Comments to the Author**

1. If the authors have adequately addressed your comments raised in a previous round of review and you feel that this manuscript is now acceptable for publication, you may indicate that here to bypass the “Comments to the Author” section, enter your conflict of interest statement in the “Confidential to Editor” section, and submit your "Accept" recommendation.

Reviewer #2: All comments have been addressed

Reviewer #3: (No Response)

2. Is the manuscript technically sound, and do the data support the conclusions?

Reviewer #2: Partly

Reviewer #3: Yes

3. Has the statistical analysis been performed appropriately and rigorously? 

Reviewer #2: Yes

Reviewer #3: Yes

4. Have the authors made all data underlying the findings in their manuscript fully available?

Reviewer #2: Yes

Reviewer #3: Yes

5. Is the manuscript presented in an intelligible fashion and written in standard English?

Reviewer #2: Yes

Reviewer #3: Yes

6. Review Comments to the Author

Reviewer #2: (No Response)

Reviewer #3: This is an interesting paper. The authors have correctly addressed most of the comments raised by reviewers #1 and #2. I have just a few comments.

Comment #1: The title is not correct. You saw the effects are ASSOCIATED with amygdalar activation, not VIA amygdala. You don´t know if other brain areas were also strongly activated. So, please change the title to: "Nerve injury induces pain hypersensitivity and anxiety-related behaviours and is associated with amygdala activation in male mice"

Other typos/mistakes to be corrected:

- line 37: "... experience accompanying egative" - "negative" is missing an "n"

- lines 63-68: this sentence has 6 lines. Please break it in two sentences

- Please improve the English of the sentence: "However, following the neuropathic pain induced by SNI, how rostro-caudal involvement of BLA and CeA remains to be unclear."

- line 89 "...Consent to Participate" - this is not a human study.

- lines 158-160: the sentence "respectively. mixed repeated ANOVA was used to campare the espression of c-Fos across different bregma distance. A significance level of p < 0.05 was considered statistically significant." has several typos (". mixed", "campare", "espression" - correct them.

- line 383: "(C) T The c-Fos expression" - there is an extra "T"

7. PLOS authors have the option to publish the peer review history of their article (what does this mean? ). If published, this will include your full peer review and any attached files.

**Do you want your identity to be public for this peer review?** For information about this choice, including consent withdrawal, please see our Privacy Policy .

Reviewer #2: No

Reviewer #3: No

---

## [Author Response · Author response to Decision Letter 2]

1 Apr 2025

Reviewer #3:

Comment #1: The title is not correct. You saw the effects are ASSOCIATED with amygdalar activation, not VIA amygdala. You don´t know if other brain areas were also strongly activated. So, please change the title to: "Nerve injury induces pain hypersensitivity and anxiety-related behaviours and is associated with amygdala activation in male mice"

Response: We sincerely appreciate your careful reading and constructive suggestion. We fully agree that the term "via amygdala activation" in the original title could imply a causal or exclusive mechanism, which is beyond the scope of our current findings. As suggested, we have revised the title to "Nerve injury induces pain hypersensitivity and anxiety-related behaviours and is associated with amygdala activation in male mice" to more accurately reflect the correlational nature of our observations. This change has been implemented in the revised manuscript (page 1 and 2, title section).

Other typos/mistakes to be corrected:

1. - line 37: "... experience accompanying egative" - "negative" is missing an "n"

Response: Thank you for your meticulous attention to detail. This typographical error has been corrected to "negative" in line 37 of the revised manuscript (page 3, line 29 ). We have also conducted a full proofread of the text to ensure no similar errors remain.

2.- lines 63-68: this sentence has 6 lines. Please break it in two sentences

Response: Thank you for identifying this overly complex sentence. We have restructured the original sentence (now lines 60-68 in the revised manuscript, page 4) to improve readability and logical flow. The revised text reads:

The amygdala, as a structurally complex and functionally diverse brain region, exhibits distinct anatomical structures and functional subdivisions across different dimensions. Through in-depth exploration, its fine architecture can be analyzed from multiple perspectives, which delineate the amygdala's internal organization. These multifaceted investigations further provide critical insights into its involvement in higher cognitive functions, particularly emotional processing, memory consolidation, and social behavior.

Additionally, we have reviewed other long sentences in the manuscript to ensure clarity and conciseness.

3.- Please improve the English of the sentence: "However, following the neuropathic pain induced by SNI, how rostro-caudal involvement of BLA and CeA remains to be unclear."

Response: We appreciate your suggestion to enhance clarity. The original sentence has been revised to improve grammatical structure and precision. The updated version in lines 66-68 (page 4) now reads:

"However, the differential involvement of BLA and CeA along their rostrocaudal axis in SNI-induced neuropathic pain remain to be fully elucidated."

4.- line 89 "...Consent to Participate" - this is not a human study.

Response: We sincerely apologise for this oversight. As this is an animal study, the phrase "Consent to Participate" has been removed from the Ethics Statement in line 89 (page 5).

5.- lines 158-160: the sentence "respectively. mixed repeated ANOVA was used to campare the espression of c-Fos across different bregma distance. A significance level of p < 0.05 was considered statistically significant." has several typos (". mixed", "campare", "espression" - correct them.

Response: We deeply appreciate your meticulous proofreading. The errors have been corrected as follows in lines 168-169 (page 8, Methods section):

". mixed" → ". Mixed".

"campare" → "compare".

"espression" → "expression".

6.- line 383: "(C) T The c-Fos expression" - there is an extra "T"

Response: Thank you for catching this formatting error. The redundant "T" in the subfigure label of Figure 4C has been removed (now line 429, page 17).

---

## [Editor Report · Decision Letter 3]

4 Apr 2025

Nerve injury induces pain hypersensitivity and anxiety-related behaviours and is associated with amygdala activation in male mice

PONE-D-24-27569R3

Dear Dr. Wu,

We’re pleased to inform you that your manuscript has been judged scientifically suitable for publication and will be formally accepted for publication once it meets all outstanding technical requirements.

Kind regards,

Armando Almeida

Academic Editor

PLOS ONE
---

## [Editor Report · Acceptance letter]

PONE-D-24-27569R3

PLOS ONE

Dear Dr. Wu,

I'm pleased to inform you that your manuscript has been deemed suitable for publication in PLOS ONE. Congratulations! Your manuscript is now being handed over to our production team.

Kind regards,

on behalf of

Prof. Armando Almeida

Academic Editor

PLOS ONE